# Evaluation of Systemic Risk Factors in Patients with Diabetes Mellitus for Detecting Diabetic Retinopathy with Random Forest Classification Model

**DOI:** 10.3390/diagnostics14161765

**Published:** 2024-08-13

**Authors:** Ramesh Venkatesh, Priyanka Gandhi, Ayushi Choudhary, Rupal Kathare, Jay Chhablani, Vishma Prabhu, Snehal Bavaskar, Prathiba Hande, Rohit Shetty, Nikitha Gurram Reddy, Padmaja Kumari Rani, Naresh Kumar Yadav

**Affiliations:** 1Department of Retina and Vitreous, Narayana Nethralaya, Bengaluru 560010, India; phgandhi28@gmail.com (P.G.); ayushichoudhary10@gmail.com (A.C.); drrupalkathare@gmail.com (R.K.); dr.vishmaprabhu@gmail.com (V.P.); snehal1128@gmail.com (S.B.); prathibahande@gmail.com (P.H.); vasudha.naresh@gmail.com (N.K.Y.); 2Medical Retina and Vitreoretinal Surgery, University of Pittsburgh School of Medicine, Pittsburg, PA 15213, USA; jay.chhablani@gmail.com; 3Department of Cornea and Refractive Services, Narayana Nethralaya, Bengaluru 560010, India; drrohitshetty@yahoo.com; 4Anant Bajaj Retina Institute, L V Prasad Eye Institute, Kallam Anji Reddy Campus, Hyderabad 500034, India; nikithareddy67345@gmail.com (N.G.R.); rpk@lvpei.org (P.K.R.)

**Keywords:** new cases, diabetes, screening, diabetic retinopathy, random forest classifier

## Abstract

Background: This study aims to assess systemic risk factors in diabetes mellitus (DM) patients and predict diabetic retinopathy (DR) using a Random Forest (RF) classification model. Methods: We included DM patients presenting to the retina clinic for first-time DR screening. Data on age, gender, diabetes type, treatment history, DM control status, family history, pregnancy history, and systemic comorbidities were collected. DR and sight-threatening DR (STDR) were diagnosed via a dilated fundus examination. The dataset was split 80:20 into training and testing sets. The RF model was trained to detect DR and STDR separately, and its performance was evaluated using misclassification rates, sensitivity, and specificity. Results: Data from 1416 DM patients were analyzed. The RF model was trained on 1132 (80%) patients. The misclassification rates were 0% for DR and ~20% for STDR in the training set. External testing on 284 (20%) patients showed 100% accuracy, sensitivity, and specificity for DR detection. For STDR, the model achieved 76% (95% CI-70.7%–80.7%) accuracy, 53% (95% CI-39.2%–66.6%) sensitivity, and 80% (95% CI-74.6%–84.7%) specificity. Conclusions: The RF model effectively predicts DR in DM patients using systemic risk factors, potentially reducing unnecessary referrals for DR screening. However, further validation with diverse datasets is necessary to establish its reliability for clinical use.

## 1. Introduction

Diabetes mellitus (DM) is a global epidemic that causes a wide range of complications in the human body [1]. According to recently released information from the International Diabetes Federation Diabetes Atlas in 2021, India has the world’s second-highest number of people with diabetes, trailing only China [2]. Diabetic retinopathy (DR) is one of the many serious ocular complications of DM [3,4]. The primary goal of screening DM patients for DR is to identify and treat sight-threatening DR [STDR], i.e., proliferative DR and/or diabetic macular edema, while also recommend follow-up for those who do not have DR or STDR [5,6]. Worldwide, existing DR screening practices include dilated fundoscopy and evaluation by ophthalmologists or teleophthalmology tools such as mydriatic and non-mydriatic fundus cameras [7]. In developing countries such as India, the current DR screening practice generally involves a dilated fundus examination by a trained retina specialist of referred DM patients identified by other specialists or general ophthalmologists, optometrists, and diabetologists. The most recent publication from the SMART India Study group revealed a national prevalence of 12.5% for DR and 4.0% for STDR [6,8]. With a large population of DM patients to be screened for DR, a low prevalence of DR and STDR, and a limited number of retina specialists for a country with high population density such as India, alternative strategies assisted by technology and artificial intelligence (AI) must be devised to screen a large number of DM patients and detect cases which have a high probability of having DR and only refer cases with STDR to retina specialists who require immediate attention. An AI-based machine learning-driven predictive model could provide a rapid and a practical solution for screening DR.

Machine learning (ML) is a branch of AI and is highly effective in creating predictive models. It learns from data without direct programming, indicating that the expression of a particular task improves with incremental data and variables. Recently, ML has demonstrated great potential for diagnostic applications [9]. Many studies in the field of AI in DR focus on utilizing color fundus and fluorescein angiography images and advanced, complicated ML techniques to identify cases of referable and non-referable DR [10,11,12,13,14,15,16,17,18,19,20,21,22,23]. To the best of our knowledge, we did not come across literature that used the patient’s demographic and systemic risk factor data to objectively predict the risk of developing DR or STDR. We propose that utilizing a patient’s demographic profile and systemic risk factors is a simple and practical approach to identifying those at high or low risk of developing DR. This method could significantly streamline DR screening processes. In a recent publication involving multiple centers, significant systemic risk factors were identified that could be utilized to determine the optimal timing for initial DR screening for maximizing the detection of DR and STDR cases. These risk factors were identified by a joint team of trained retina experts and multiple text-based generative AI sources [24]. With this knowledge, we aimed to develop a predictive model using supervised ML, trained on patients’ demographic and systemic risk factor data, to accurately predict the likelihood of DR at initial screening. This model could be a valuable tool for healthcare professionals, including non-retina specialists, general ophthalmologists, optometrists, and diabetologists, helping them determine which DM patients need direct screening by a retina specialist and which can be screened via teleophthalmology without direct referral.

In the field of data mining for disease prediction, a range of supervised ML algorithms have been employed [25]. The Support Vector Machine model was commonly employed; however, the Random Forest (RF) model exhibited higher accuracy due to its ability to minimize bias and variance [25].

Thus, the main objective of this study was to evaluate the predictive efficacy of a supervised ML-based RF classifier model for detecting DR based on the patient’s demographic and systemic risk factor information at a hospital-based setting.

## 2. Methods

### 2.1. Dataset

This retrospective cohort study included patients aged ≥18 years diagnosed with type 1 or type 2 DM who visited our hospital’s diabetic and/or retina clinics for the first time for DR screenings between January and December 2023. Demographic and medical history data were extracted from electronic medical records and documented in Microsoft^®^ Excel 2021. Collected information included age, gender, DM type, treatment history (oral hypoglycemic agents, insulin, or both), DM control (HbA1C levels if available), family history of DM, pregnancy status during DR screening, and associated comorbidities (hypertension, renal disease, cerebrovascular accident, cardiovascular disease, dyslipidemia, and anemia).

A trained retina specialist (RV) with over 10 years of experience conducted fundus evaluations using indirect ophthalmoscopy under mydriasis and documented findings with Ultrawide field Optos^®^ imaging (Optos, Daytona, UK). Patients were diagnosed with DR if any signs of DR were present in either eye, irrespective of severity, stage, or macular status. The presence of STDR was noted based on signs of proliferative DR, diabetic macular edema, or both in any eye. Missing values were estimated using mean imputation for quantitative variables and mode imputation for qualitative variables in Microsoft^®^ Excel with XLSTAT (v2023.3.1.1416) (Denver, CO, USA).

### 2.2. Computational Methods

The collected dataset was divided into training (80%) and testing (20%) sets. An ensemble learning approach using the RF classifier algorithm was employed. The training samples were chosen randomly with replacement, and a bagging technique was used to create a forest of 500 trees, each consisting of 300 samples. The final prediction was derived from the majority vote of the individual decision trees. The model parameters were optimized to ensure minimal mean square error and high prediction accuracy. Parameters included a minimum node size of 2, minimum child node size of 1, maximum tree depth of 20, and a complexity parameter of 0.0001. These values were determined through various permutations and combinations to balance model complexity and generalization, thereby mitigating overfitting while ensuring accurate predictions.

Two RF classifier models were developed, each with distinct dependent variables. The first model (RF1) was designed to detect any stage of DR, while the second model (RF2) focused on detecting STDR. Both models utilized the same set of independent variables. The RF models employed out-of-bag (OOB) evaluation for internal cross-validation, treating the training set as a test set for quality assessment. The trained RF1 and RF2 models were then used to predict the likelihood of DR and STDR, respectively, on an independent test dataset that had not been used in training to ensure unbiased results. The classifications generated by the models for the test set were assessed for their effectiveness using the sensitivity analysis in the XLSTAT statistical software (v2023.3.1.1416) (Figure 1).

## 3. Results

The dataset included systemic risk factor information from 1416 DM patients. The data were split into training (80%, 1132 patient samples) and testing (20%, 284 patient samples) sets. Table 1 summarizes the patient information in both sets, providing an overview of their distribution and representation.

The model’s accuracy was assessed by quantifying the misclassification rate for the OOB evaluation samples. The study found correct classification rates of 100% for detecting any DR and approximately 80% for detecting STDR in validation samples. For OOB samples, the positive and negative predictive values were 100% for DR and 85% and 64% for STDR, respectively. The RF models identified family history of diabetes mellitus as the most significant predictor for detecting both DR and STDR stages (Table 2).

Table 3 represents a contingency table comparing the RF model predictions with clinician observations for each test sample. Sensitivity analyses results are summarized in Table 4. For test samples, the RF model detected DR with 100% accuracy (95% CI: 100%–100%), 100% sensitivity (95% CI: 100%–100%), and 100% specificity (95% CI: 100%–100%). For STDR, the model achieved a 76% accuracy (95% CI: 70.7%–80.7%), 53% sensitivity (95% CI: 39.2%–66.6%), and 80% specificity (95% CI: 74.6%–84.7%).

## 4. Discussion

In summary, this study utilized the demographic and systemic medical history data of DM patients to develop a supervised ML-based predictive model. This model was capable of detecting and classifying patients into two groups: those with the potential to develop DR or STDR, and those without. The model demonstrated a remarkable degree of accuracy for predicting DR in the study. Furthermore, when evaluating the model using new patient samples, it exhibited a 100% accuracy rate for detecting DR and a 76% accuracy rate for detecting STDR.

Various risk factors influence DR screening in newly diagnosed DM patients, either independently or interdependently. The American Diabetes Association (ADA) guidelines, widely accepted worldwide, primarily base DR screening recommendations on two risk factors: type of DM and pregnancy status [5]. However, community-based studies have identified at least 12 risk factors that affect the development or progression of DR, making the ADA guidelines appear overly simplistic and outdated [26]. Therefore, developing a concise set of key risk factors for routine clinical use has become essential.

In a recent multicenter study involving our group, relevant risk factors of varying importance were identified to optimize the timing of initial DR screening, maximizing the yield of DR and STDR cases for retina specialists. A collaborative team of retina experts and text-based generative AI resources identified these risk factors [24]. Consequently, this study developed a prediction model for detecting DR and STDR using the identified risk factors and patient demographic data.

One notable exclusion from our study’s pool of risk factors for identifying DR was the duration of DR. We are aware that DR is a retinal complication of long-term DM that affects the retinal microvascular system [27]. As a result, the risk of developing DR or STDR increases with longer duration of DM. However, in real-world clinical practice, particularly in low- or middle-income countries where annual screening for DM is not routinely performed, knowing the exact duration of DM is unlikely. Even the ADA guidelines advise patients with type 2 DM to be screened promptly after diagnosis because many patients with type 2 DM have the disease for a long time before being diagnosed. Patients with type 1 DM, on the other hand, are advised to undergo screening within 5 years from diagnosis [5]. Thus, the duration of DM does not appear to be an important risk factor when screening for DR.

As per the ADA, AI has the potential to serve as a substitute for conventional screening techniques in identifying DR [28]. Nonetheless, the use of AI is not recommended for patients with a history of DR, previous DR treatment, or any signs of vision impairment. Recent studies have explored various deep learning techniques for DR classification, showcasing significant advancements in this field [10,11,12,13,14,15,16,17,18,19,20,21,22,23,29]. One approach utilized convolutional neural networks (CNNs) such as VGG16 and VGG19, achieving an accuracy of 90.60% and a 94% F1 score by classifying DR into five severity levels across multiple datasets, including APTOS-2019 and Messidor-2 [15]. Another study by Gunasekaran et al. employed a deep recurrent neural network (RNN) for DR prediction from fundus images, attaining a precision of 95.5%. Khan et al. compared several deep neural network architectures, with InceptionV3 showing the highest testing accuracy of 79.4% [16]. Fang et al. introduced a Directed Acyclic Graph (DAG) network model for multi-feature fusion, evaluated on hospital and DIARETDB1 datasets [17]. Elloumi et al. addressed the challenge of smartphone-captured fundus images using NasnetMobile, achieving a 95.91% accuracy [30]. Kanakaprabha et al. [18] assessed various architectures, including VGG16 and ResNet50, for DR prediction. Sridhar used a CNN to classify DR severity with notable accuracy improvements on a Kaggle dataset, while Das et al. applied morphology and adaptive histogram equalization with a CNN, achieving 97.2% precision [19]. Vives-Boix et al. [31] implemented meta-plasticity in CNNs, achieving a 95.56% accuracy, and Adriman evaluated ResNet and DenseNet, with ResNet reaching 96.25% accuracy [32]. Fatima’s hybrid neural network model demonstrated a strong performance on the MESSIDOR-2 and APTOS datasets, and Qureshi’s ADL-CNN model achieved a 98.0% accuracy [22,33]. Kalyani et al. applied capsule networks, achieving a high accuracy across DR stages [34]. Gayathri’s multipath CNN combined with ML classifiers showed strong results on IDRiD and MESSIDOR datasets [20]. Bodapati’s composite DNN with a gated-attention mechanism performed well on the APTOS-2019 dataset, and Math’s system based on a pre-trained CNN achieved 96.37% sensitivity and specificity [21,22]. Gao’s grading system using fundus fluorescein angiography images and deep learning algorithms achieved 94.17% accuracy, while Kobat’s DenseNET model, using horizontal and vertical image patches, achieved 84.90% accuracy with 10-fold cross-validation [23,35]. These studies collectively highlight the effectiveness of various deep learning methods for DR classification, reflecting high accuracy and promising results across diverse datasets. However, there are valid concerns regarding image acquisition and quality, potential biases in the data and the determination of ground truth, the selection of appropriate algorithms, challenges associated with deep ML, the applicability of AI in diverse populations, and the obstacles faced in the adoption of AI in healthcare [36]. To address this issue, a powerful predictive AI model using ML which would utilize patient information, like demographic details and medical history, rather than relying on fundus images alone is required. The RF model is a powerful ML method that utilizes ensemble learning techniques to address classification and regression problems. It creates multiple decision trees during training and outputs the most frequent class (classification) or the mean prediction (regression) from these trees. The RF classifier model offers a high accuracy by reducing biases and variances commonly observed in single decision tree models. Therefore, we used the RF classifier model to train and test individual patient samples for identifying DR in this study.

The study achieved correct classification rates of 100% for DR and ~80% for STDR in the training samples. For the test samples, the RF model detected DR with 100% accuracy and STDR with 76% accuracy, indicating very high training accuracy and low variance between the training and testing sets. Although a 100% accuracy rate for DR detection might suggest overfitting, overfitting is characterized by a low number of training errors and a high number of testing errors. In this study, the numbers of both training and testing errors were low, indicating low bias and low variance. This demonstrates that the model was not overfitted but rather balanced, achieving high accuracy and repeatability in detecting DR and STDR.

This study has several positive implications for clinical practice in DR screening for newly diagnosed DM cases. First, integrating this model into hospital electronic medical records will enable general ophthalmologists, non-retina specialists, optometrists, diabetologists, and general physicians to identify which DM cases require direct referral to retina specialists for DR screening via dilated fundus examination. Second, by classifying DR and STDR, patients without STDR can be screened using alternative tools such as teleophthalmology with mydriatic and non-mydriatic fundus cameras. Third, this approach can reduce the burden on retina specialists by involving more healthcare providers in DR screening. Fourth, by referring only high-risk DM cases for treatment-requiring DR, retina specialists can focus on cases needing immediate attention, enhancing their efficiency. Finally, from a patient perspective, it can reduce hospital or clinic wait times.

This study has some limitations. The size of the dataset was relatively small given the high prevalence of DM in the population. The study samples were not compared with other supervised ML algorithms. The use of convenience sampling, relying on DM cases referred to an eye hospital for DR screening, may have introduced sample bias. To enhance the real-world applicability of the model, additional validation in varied settings, such as community health programs or screening camps, is necessary. Moreover, the research did not study which systemic factors pose the greatest risk for developing DR, which could aid clinicians in identifying high-risk patients before screening. However, the primary aim was to develop a straightforward and accurate model for identifying and categorizing DR without relying on fundus images, thereby minimizing unnecessary referrals to retina specialists. To enhance the model’s reliability and acceptability in clinical practice, further research is needed with training and testing samples from diverse cohorts. This future research should objectively assess the model’s clinical impact. Once validated, the model can be integrated into hospital electronic medical records as a viable alternative screening tool for DR.

In conclusion, the ML-based RF classifier model shows promise in detecting and classifying DR and STDR in diagnosed DM patients screened for the first-time, potentially transforming DR screening practices. However, further validation studies are necessary to confirm its reliability and suitability as an alternative method for predicting DR.

## Figures and Tables

**Figure 1 diagnostics-14-01765-f001:**
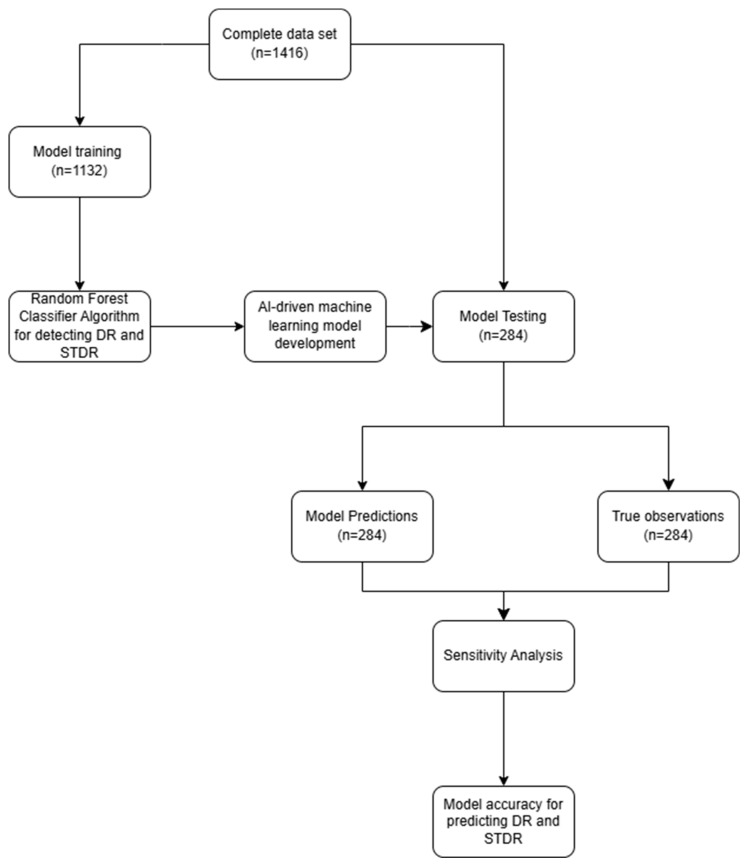
Flow chart depicting the process of model training and testing in the study.

**Table 1 diagnostics-14-01765-t001:** Distribution and representation of different sample categories of patients with diabetes mellitus used for this study.

Variables		Model Training	Model Testing
Sample size (*n*, %)		1132 (80)	284 (20)
Age (mean ± SD)		49.28 ± 13.90	46.55 14.1
Gender	Female (*n*, %)	390	91 (32)
	Male (*n*, %)	742	193 (68)
Type of DM	T1DM (*n*, %)	39 (3)	13 (5)
	T2DM (*n*, %)	1093 (97)	271 (95)
HbA1C levels (mean ± SD)		8.93 ± 2.12	9.26 1.42
DM treatment with	No treatment (*n*, %)	55 (5)	14 (5)
	OHA (*n*, %)	813 (72)	212 (75)
	Insulin (*n*, %)	79 (7)	19 (7)
	OHA + Insulin (*n*, %)	185 (16)	39 (14)
Control of DM	Controlled (*n*, %)	551 (49)	141 (50)
	Not controlled (*n*, %)	581 (51)	143 (50)
Family history of DM	Absent (*n*, %)	569 (50)	138 (49)
	Present (*n*, %)	563 (50)	146 (51)
Pregnancy status	Not pregnant (*n*, %)	1132 (100)	284 (100)
Systemic co-morbidity	Absent (*n*, %)	598 (53)	160 (56)
	Present (*n*, %)	534 (47)	124 (44)
Presence of DR in any eye	Absent (*n*, %)	569 (50)	138 (49)
	Present (*n*, %)	563 (50)	146 (51)
Presence of STDR in any eye	Absent (*n*, %)	828 (73)	212 (75)
	Present (*n*, %)	304 (27)	72 (25)

Abbreviations: DM—diabetes mellitus; DR—diabetic retinopathy; STDR—sight threatening diabetic retinopathy; OHA—oral hypoglycemic agents; SD—standard deviation; T1DM—Type 1 diabetes mellitus; T2DM—Type 2 diabetes mellitus.

**Table 2 diagnostics-14-01765-t002:** Variable importance for detecting DR and STDR using Random Forest Classifier Model.

Factors	Any DR Stage			STDR Stage		
	Absent	Present	Overall	Absent	Present	Overall
Age	1.567	4.495	4.332	4.848	1.850	4.798
Gender	0.286	0.645	0.656	1.190	0.419	1.027
HbA1c	0.700	4.714	4.684	1.755	−1.966	−0.139
DM type	2.157	1.587	3.327	−0.711	0.810	0.103
DM treatment	5.537	3.493	6.042	−0.264	2.581	1.997
DM control	0.311	1.998	2.103	1.963	−1.251	0.959
Family history of DM	99.884	112.021	121.556	27.086	45.613	41.069
Pregnancy status	0.000	0.000	0.000	0.000	0.000	0.000
Systemic co-morbidity	1.384	1.410	1.846	−0.364	−0.250	−0.535

Abbreviations: DR—diabetic retinopathy; STDR—sight threatening diabetic retinopathy; HbA1c—Glycosylated hemoglobin; DM—diabetes mellitus.

**Table 3 diagnostics-14-01765-t003:** Contingency tables comparing the model’s predictions against the clinician’s observations.

Observed Data (*n* = 284)	Model Prediction (*n* = 284)		Observed Data (*n* = 284)	Model Prediction (*n* = 284)	
	DR Present (*n*, %)	DR Absent (*n*, %)		DR Present (*n*, %)	DR Absent (*n*, %)
DR Present (*n*, %)	146 (51)	0 (0)	STDR Present (*n*, %)	25 (9)	47 (16)
DR Absent (*n*, %)	0 (0)	138 (49)	STDR Absent (*n*, %)	22 (8)	190 (67)

Abbreviations: DR—diabetic retinopathy; STDR—sight threatening diabetic retinopathy.

**Table 4 diagnostics-14-01765-t004:** Sensitivity analyses of the testing samples for detecting DR and STDR.

Statistic	Analysis for Detecting DR			Analysis for Detecting STDR		
	Value	Lower Bound (95%)	Upper Bound (95%)	Value	Lower Bound (95%)	Upper Bound (95%)
Correct classification	1.000	1.000	1.000	0.757	0.707	0.807
Misclassification	0.000	0.000	0.000	0.243	0.193	0.293
Sensitivity	1.000	0.968	1.000	0.532	0.392	0.666
Specificity	1.000	0.967	1.000	0.802	0.746	0.847
False positive rate	0.000	0.000	0.000	0.198	0.148	0.249
False negative rate	0.000	0.000	0.000	0.468	0.331	0.605
Prevalence	0.514	0.456	0.572	0.165	0.122	0.209
Positive Predictive Value	1.000	1.000	1.000	0.347	0.237	0.457
Negative Predictive Value	1.000	1.000	1.000	0.896	0.855	0.937

Abbreviations: DR—diabetic retinopathy; STDR—sight threatening diabetic retinopathy.

## Data Availability

The datasets generated during and/or analyzed during the current study are available from the corresponding author on reasonable request.

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
