# Peer review of "Evaluation of Systemic Risk Factors in Patients with Diabetes Mellitus for Detecting Diabetic Retinopathy with Random Forest Classification Model"

_diagnostics, 2024, doi:10.3390/diagnostics14161765_

Round 1

Reviewer 1 Report

Comments and Suggestions for Authors

The abstract has inconsistent use of punctuation and sentence structure. For example, “To evaluate systemic risk factors in diabetes mellitus (DM) patients to attempt to predict diabetic retinopathy (DR) using Random Forest (RF) classification model.” could be rephrased for clarity.

Some sentences are fragmented or awkwardly constructed, such as "The model's sensitivity and specificity were determined by testing it on a separate dataset." This could be simplified or expanded for clarity.

The results section mentions “100% accuracy” multiple times without providing a context or explanation about the dataset's nature, which might mislead readers about the model's real-world applicability.

Also, based on what you did for overfitting, the model appears to have been overfitted.

Why were these algorithms used? What is their superiority? Check it out in the article.

The description of the Random Forest model parameters and construction is somewhat vague. More details on how the parameters were chosen would enhance reproducibility.

The explanation of the dataset splitting into training, validation, and test sets lacks clarity.

There is a lack of a comprehensive literature review on similar studies that have used machine learning for predicting diabetic retinopathy. Including a comparison with previous works would provide context and justification for this study.

The methodology section should include more details on the preprocessing steps for the data, such as handling missing values ​​and the specific techniques used.

The paper should describe how the dataset was balanced if it was imbalanced, as this is crucial for classification tasks.

The discussion section does not adequately address the limitations of the study. For example, the potential overfitting due to high accuracy on the training set is not discussed.

There is no mention of future work or how this model could be improved or validated with different datasets.

Comments on the Quality of English Language

minor editing

Author Response

  • Reviewer 1:

    The abstract has inconsistent use of punctuation and sentence structure. For example, “To evaluate systemic risk factors in diabetes mellitus (DM) patients to attempt to predict diabetic retinopathy (DR) using Random Forest (RF) classification model.” could be rephrased for clarity.

    Reply: At the outset, we thank the reviewer for reviewing the manuscript and providing us with some valuable feedbacks for improvising it.

    Thank you for pointing out the errors related to grammar and punctuation. We have rectified this in the entire manuscript to the best of our ability.

    Some sentences are fragmented or awkwardly constructed, such as "The model's sensitivity and specificity were determined by testing it on a separate dataset." This could be simplified or expanded for clarity.

    Reply: Thank you for pointing this out. Changes have been made in the revised section of the manuscript.

    The results section mentions “100% accuracy” multiple times without providing a context or explanation about the dataset's nature, which might mislead readers about the model's real-world applicability.

    Also, based on what you did for overfitting, the model appears to have been overfitted.

    Reply: For a model to be considered for overfitted, there needs to be a low bias (low training error) and high variance (high testing error). The model clearly suggested that the training error of the model was and the predictions for the testing samples were low as well. This clearly demonstrates that the model was not overfitted but rather balanced, achieving high accuracy and repeatability in detecting DR and STDR. This point related to overfitting has been mentioned in detail in the revised manuscript and has been marked in red.

    Why were these algorithms used? What is their superiority? Check it out in the article.

    Reply: In this study, we directly selected the Random Forest Model rather than training and comparing the data with the different supervised machine leaning models. This is definitely one of the drawbacks of the study and this has been mentioned in the limitations section of the manuscript. Based on the previous literature, and the model’s ability to tackle issues related to over or under fitting, we directly selected the RF model over other ML models.   

    The description of the Random Forest model parameters and construction is somewhat vague. More details on how the parameters were chosen would enhance reproducibility.

    The explanation of the dataset splitting into training, validation, and test sets lacks clarity.

    Reply: We agree with reviewer that the technique of presentation of the data in the initial version of the manuscript was complex. We have taken steps to simplify the methodology in the revised version of the manuscript. We have divided the methodology into two sections: 1) description of the data set and 2) description of the computational method. All these changes are represented in the revised version of the manuscript and has been marked in red.

    There is a lack of a comprehensive literature review on similar studies that have used machine learning for predicting diabetic retinopathy. Including a comparison with previous works would provide context and justification for this study.

    Reply: We have included a literature review of similar studies relevant to the topic and included them in the discussion section of the revised version of the manuscript.  

    The methodology section should include more details on the preprocessing steps for the data, such as handling missing values ​​and the specific techniques used.

    The paper should describe how the dataset was balanced if it was imbalanced, as this is crucial for classification tasks.

    Reply: This point has been included and addressed in the methodology section of the revised manuscript and is marked in red.

    The discussion section does not adequately address the limitations of the study. For example, the potential overfitting due to high accuracy on the training set is not discussed.

    Reply: The point related to the overfitting of the model has been adequately addressed in the discussion section of the revised manuscript.   

    There is no mention of future work or how this model could be improved or validated with different datasets.

    Reply: This point has been included in the revised manuscript.

Reviewer 2 Report

Comments and Suggestions for Authors

The topic of the article is sound and important from a scientific and clinical point of view. Some solutions are already in place, but this requires further research. The quality of the article would be enhanced by the following amendments:

1. Dividing section 2 Material and Methods into two subsections 2.1 Data set and 2.2 Computational methods would improve the clarity of the article.

2. Justification for the choice of RF and STDR among the many possible methods of analysis seems necessary.

3. From what was the number of patients in the dataset and the number of features? Convenience sample?

4. Has an attempt been made to cross-calidate the methods used?
5. The discussion should be divided into a number of sub-sections: results of previous research and their limitations and how to overcome them in the self-study, limitations of the self-study, key directions for further research.

6. The literature review is too poor - 478 publications were found in PubMed with the keywords "diabetes "+"retinopathy "+"mechine learning" (published 2007-2024).

Author Response

Reviewer 2

The topic of the article is sound and important from a scientific and clinical point of view. Some solutions are already in place, but this requires further research.

Reply: At the outset, we thank the reviewer for reviewing the manuscript and providing us with some valuable feedbacks for improvising it.

The quality of the article would be enhanced by the following amendments:

  • Dividing section 2 Material and Methods into two subsections 2.1 Data set and 2.2 Computational methods would improve the clarity of the article.

Reply: As suggested by the reviewer, we have included these two sections in the revised manuscript. Please see the methodology section of the revised manuscript.

  • Justification for the choice of RF and STDR among the many possible methods of analysis seems necessary.

Reply: Justification regarding the choice of RF for DR and STDR detection has been explained in the introduction and the discussion section of the manuscript and has been marked in red.

  • From what was the number of patients in the dataset and the number of features? Convenience sample?

Reply: The point of convenience sampling has been addressed in the limitation section of the revised manuscript.

  • Has an attempt been made to cross-validate the methods used?

Reply: No. Cross validation between the different ML methods was not done in this study. Reply: In this study, we directly selected the Random Forest Model rather than training and comparing the data with the different supervised machine leaning models. This is definitely one of the drawbacks of the study and this has been mentioned in the limitations section of the manuscript. Based on the previous literature, and the model’s ability to tackle issues related to over or under fitting, we directly selected the RF model over other ML models.

  • The discussion should be divided into a number of sub-sections: results of previous research and their limitations and how to overcome them in the self-study, limitations of the self-study, key directions for further research.

Reply: This has been done as suggested by the reviewer. Please see the discussion section of the revised manuscript.

  • The literature review is too poor - 478 publications were found in PubMed with the keywords "diabetes "+"retinopathy "+"machine learning" (published 2007-2024).

Reply: A discussion on the previous studies published on this topic has been included in the discussion section of the revised manuscript.

Reviewer 3 Report

Comments and Suggestions for Authors

Hermosillo, July  31st, 2024

Evaluation of systemic risk factors in patients with diabetes mellitus for detecting diabetic retinopathy with Random Forest classification model.

Diabetes mellitus (DM) is a global epidemic that causes a wide range of complications in the human body. According to information recently published by the International Diabetes Federation's Diabetes Atlas in 2021, India has the second highest number of people with diabetes in the world, behind only China. Diabetic retinopathy (DR) is one of the many serious ocular complications of DM.

The topic presented by the authors is very interesting. Furthermore, the authors highlight the importance of the impact of the evaluation of systemic risk factors in patients with diabetes mellitus for the detection of diabetic retinopathy with the Random Forest classification model. 

After reading the complete manuscript, I have observed that the literature consulted for this manuscript is well consulted and updated, but I have some questions for the authors of the work, which are the following:

1. With the use of Random Forest (RF) classification. What are the systemic risk factors that differentiate diabetic retinopathy (DR) and treat sight-threatening retinopathy [STDR]?

2. Are there any differences in factors such as patient age, sex, type of diabetes (type 1 or 2) in DR and STDR?

3. Could you explain the mechanism in eye diseases such as diabetic retinopathy, chronic anterior uveitis, glaucoma, dry eye syndrome and AMD?

4. What recommendation do you have for ophthalmologists, optometrists or diabetologists for patients with diabetes mellitus and diabetic retinopathy?

Author Response

Reviewer 3:

Evaluation of systemic risk factors in patients with diabetes mellitus for detecting diabetic retinopathy with Random Forest classification model.

Diabetes mellitus (DM) is a global epidemic that causes a wide range of complications in the human body. According to information recently published by the International Diabetes Federation's Diabetes Atlas in 2021, India has the second highest number of people with diabetes in the world, behind only China. Diabetic retinopathy (DR) is one of the many serious ocular complications of DM.

The topic presented by the authors is very interesting. Furthermore, the authors highlight the importance of the impact of the evaluation of systemic risk factors in patients with diabetes mellitus for the detection of diabetic retinopathy with the Random Forest classification model. 

After reading the complete manuscript, I have observed that the literature consulted for this manuscript is well consulted and updated, but I have some questions for the authors of the work, which are the following:

Reply: At the outset, we thank the reviewer for reviewing the manuscript and providing us with some valuable feedbacks for improvising it.

  1. With the use of Random Forest (RF) classification. What are the systemic risk factors that differentiate diabetic retinopathy (DR) and treat sight-threatening retinopathy [STDR]? Are there any differences in factors such as patient age, sex, type of diabetes (type 1 or 2) in DR and STDR?

Reply: Thank you for highlighting this crucial point. The study did not aim to identify systemic factors posing the highest risk for developing DR and STDR, which could assist clinicians in recognizing high-risk patients prior to screening. However, the RF model analysis indicated that family history of DM was the most significant predictor of DR and STDR in the training dataset, as shown in Table 2 of the revised manuscript. The primary objective of the study was to develop a simple and accurate model for identifying and categorizing DR without fundus images, thereby reducing unnecessary referrals to retina specialists.

2) Could you explain the mechanism in eye diseases such as diabetic retinopathy, chronic anterior uveitis, glaucoma, dry eye syndrome and AMD?

Reply: We failed to understand this comment and have not addressed it in the revised manuscript.

3) What recommendation do you have for ophthalmologists, optometrists or diabetologists for patients with diabetes mellitus and diabetic retinopathy?

Reply: Practical implications of the study has been mentioned in detailed in the revised section of the manuscript. By using this ML model, we would suggest that only cases prone to develop STDR need referral to retina specialists while cases without DR or without STDR can be screened with teleophthalmology tools such as mydriatic or non-mydriatic fundus photography.

Round 2

Reviewer 1 Report

Comments and Suggestions for Authors

The authors have addressed my comments and the article may be published.

Reviewer 2 Report

Comments and Suggestions for Authors

All my comments were taken into account.